

# Origin of secondary fatty alcohols in atmospheric aerosols in a cool-temperate forest based on their mass size distributions

Yuhao Cui[1,2], Eri Tachibana[2], Kimitaka Kawamura,[3] and Yuzo Miyazaki[2]

[1]Graduate School of Environmental Science, Hokkaido University, Sapporo, 060-0810, Japan
[2]Institute of Low Temperature Science, Hokkaido University, Sapporo, 060-0819, Japan
[3]Chubu Institute for Advanced Studies, Chubu University, Kasugai 487-8501, Japan

*Correspondence to*: Yuzo Miyazaki (yuzom@lowtem.hokudai.ac.jp)

**Abstract.** Fatty alcohols (FAs) are major components of surface lipids in plant leaves and serve as surface-active organic aerosols (OAs), which can act as primary biological aerosol particles (PBAPs). To elucidate the origin and formation of secondary fatty alcohols (SFAs) in atmospheric aerosols, their mass size distribution in aerosol samples obtained from a deciduous forest canopy was measured in spring, summer, and autumn. The SFAs showed the highest concentration in spring (growing season), with *n*-nonacosan-10-ol being the most abundant. In spring and summer, the size peak of *n*-

nonacosan-10-ol was in particle size range >10.0 μm, whereas it was in the 1.9–3.0 μm range in autumn. The size distribution of *n*-nonacosan-10-ol did not show any significant correlation with that of the known biogenic tracers of pollen, soil, and fungal spores in spring and summer. The overall results, together with SFAs measured in plant leaves, as well as the literature, suggest that SFAs originate mostly from plant waxes, and that leaf senescence status is likely an important factor controlling the size distribution of SFAs. This study provides new insights into the possible sources of PBAPs and their

effects on the ice nucleation activity of aerosols based on seasonal changes in particle size.

## 1 Introduction

Terrestrial lipids are key tracers for investigating the origins of atmospheric aerosols (Chen et al., 2021). Fatty alcohols (FA), with chains varying between $C_{20}$ and $C_{34}$, are major components of surface lipids (Jetter and Kunst, 2008; Kawamura et al., 2003). Once emitted into the atmosphere as primary biological aerosol particles (PBAPs), FAs can act as surface-active

organic aerosols (OAs). Among various effects, surface-active OAs can change the surface tension at air–water interfaces (Donaldson et al., 2006; You et al., 2020) and affect the atmospheric lifetime of particles (Gilman et al., 2004). In addition, FAs can nucleate ice (Perkins et al., 2020), which is important for cloud formation. Qiu et al. (2017) suggested that monolayers of *n*-alkyl alcohols, with carbon numbers up to 30, can act as efficient ice nucleants—a common ability of long-chain FAs (Vazquez de Vasquez et al., 2020). The ice nucleation properties of PBAPs (e.g. bacteria, fungal spores, and

pollen) have been widely studied (e.g. Lukas et al., 2020; O'Sullivan et al., 2015; Hader et al., 2013). However, data on the surface lipids or plant waxes from terrestrial higher plants as PBAPs are limited.

Monolayers of long chain (carbon number ≥16) linear primary alcohols (PAs) have historically been used to initiate ice nucleation (Huang et al., 2018). PAs can be oxidized to carboxylic acids, with aldehydes as intermediate components, in the aqueous phase (Yu et al., 2005). When emitted into the atmosphere, FA can be a component of the surfactant film coating on

aqueous aerosols, which affects droplet growth interactions (Li et al., 2019). Primary FAs (PFAs) are less stable in the atmosphere than secondary FAs (SFAs) because H-abstraction from secondary alcohols occurs less frequently than that from primary alcohols (Koivisto et al., 2015). SFAs are less affected by atmospheric photochemical reactions than PFAs and may, therefore, serve as tracers of specific sources. Previous studies have shown that SFAs predominate in more than 50% of the



epicuticular waxes in some plants, such as *Tropaeolum majus*, *Pinus pinaster*, *Pinus halepensis* and most conifers (e.g. Koch et al., 2006; Matas et al., 2003; Nikolić et al., 2020).

Among SFAs, *n*-nonacosan-10-ol, forming tubular aggregates (Jetter et al., 1994), is a common (in some plants, the most abundant) component of epicuticular leaf waxes (Spangenberg et al., 2010) and one of the most abundant substances in plant

tubules (Wang et al., 2015). For atmospheric aerosols, Oros and Simoneit (2001) detected *n*-nonacosan-10-ol in smoke samples from conifers subjected to controlled burning. Another SFA compound, *n*-nonacosan-5,10-ol, was identified in ambient aerosol samples collected in the western North Pacific during the Asian Pacific Regional Aerosol Characterization Experiment (ACE-Asia) campaign (Simoneit et al., 2004). Gagosian et al. (1987) emphasized that aerosol lipids can be transported from the land surface to the oceans in a relatively shorter timescale of a few days, indicating the possible impact

of SFAs on the ice nucleation (IN) activity in aerosol particles, not only on local scales but also on regional scales. Miyazaki et al. (2019) identified five SFAs in aerosols in a forest atmosphere for the first time, and showed distinct seasonal variations in their mass concentrations. The above-mentioned studies suggest that SFAs in plant leaves can potentially be emitted into the atmosphere in natural terrestrial environments and can subsequently act as PBAPs.

Despite the importance of SFAs in atmospheric aerosols in terms of their possible effects on atmospheric chemistry and

climate, studies are limited, particularly on the origins of SFAs in atmospheric aerosols, owing to a lack of aerosol samples obtained in the vicinity of possible source regions. This study was aimed at elucidating the origin and formation of SFAs based on size-segregated aerosol samples collected at a deciduous forest site during different seasons.

## 2 Experimental

### 2.1 Size-segregated aerosol sampling

Aerosol samplings were conducted at Sapporo Forest meteorology research site (42°59′ N, 141°23′ E, 182 m a.s.l.) in the western part of Hokkaido, the northernmost major island of Japan (Miyazaki et al., 2012). The sampling site (**Fig. 1**) is situated in a cool temperate zone with an annual mean temperature of 6.5°C and an annual precipitation of 1100 mm (Nakai et al., 2003). This mixed cool-temperate forest consists of a mature and secondary deciduous forest and a man-made coniferous forest with varied forest floor cover. The experimental site was covered with a broad-leaf forest consisting of

White Birch (*Betula platyphylla*), Sasa bamboo (*Sasa kurilensis*), and other species (Nakai et al., 2003). The canopy has a mean height of 20 m and forest floor is 0.5–2 m high. The evolution pattern of leaf area index (LAI) at the sampling site was similar to that at other temperate deciduous forests. The LAI determined by Nakai et al. (2003), showing seasonal variations after the initial foliating period, remained almost constant at approximately 4.0 from the end of June to mid-September. The predominant local wind direction and wind speed from May to October at this forest site indicated that the majority of

aerosols sampled was likely influenced by emissions from forested areas (Miyazaki et al., 2012).

To obtain the mass-size distributions of the aerosol SFAs and related compounds, size-segregated aerosol sampling was performed using an Andersen-type cascade impactor (AN-2100, Tokyo Dylec. Corp.). The nine-stage impactor collected supermicrometre particles (with diameters ($D_p$) of 1.0–1.9 μm, 1.9–3.0 μm, 3.0–4.3 μm, 4.3–6.4 μm, 6.4–10.0 μm, and >10.0 μm) in six stages and submicrometre (with $D_p$ <0.39 μm, 0.39–0.58 μm, and 0.58–1.0 μm) in three stages. The size-

segregated aerosol samples were collected on prebaked quartz-fibre filters (ID, 8 cm) at a flow rate of 120 L min$^{-1}$ without temperature and humidity control. The impactor was placed 2 m above the forest floor. The duration of each sampling was approximately one week. Size-segregated aerosol data obtained from 26 April to 28 May in 2010; 22 June to 13 August in 2010; and 6 October to 28 October in 2009, are shown; these periods are defined as spring, summer, and autumn, respectively. The collected filter samples were stored in glass jars at −20 °C until analysis. In total, nine sets of samples were

used to measure SFAs and related compounds.



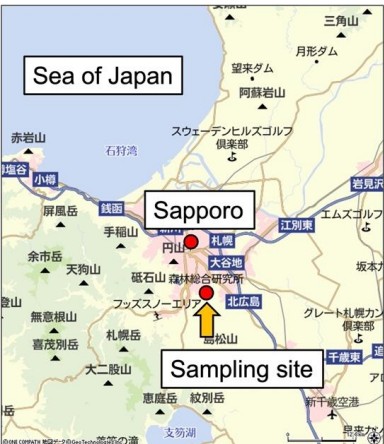

Figure 1: Location of the aerosol sampling site (copyright by GeoTechnologies).

## 2.2 Chemical analysis

To measure SFAs and related compounds, a quarter of each sample filter with an area of 10.18 cm$^2$ was extracted using a mixture of dichloromethane and methanol (2:1, v/v). The hydroxyl functional group (−OH) in extracted sample was derivatized with a mixture of 50 μL $N,O$-bis-(trimethylsilyl) trifluoroacetamide (BSTFA) and 10 μL pyridine to form trimethylsilyl (TMS) ether (−OTMS) (Fu et al., 2009). The TMS derivatives were injected into a capillary gas chromatograph (GC8890, Agilent), equipped with a fused silica capillary column (DB-5MS) and coupled to a mass spectrometer (MSD5977B, Agilent), for analysis. Total ion chromatograms (TICs) of the five derivatized SFAs were identified. The identification of each peak was based on a detailed interpretation of the EI mass spectral data (mass spectrometric fragmentation patterns) together with their comparison to data in the literature as well as exact mass measurements using a high-resolution gas chromatograph-time-of-flight mass spectrometer (GC-TOF-MS; JMS-T100GCV, JEOL) at Creative Research Institute of Hokkaido University, as described by Miyazaki et al. (2019). **Fig. 2a** shows molecular structures of five SFAs, namely $n$-heptacosan-10-ol (SFA1), $n$-heptacosan-5,10-diol (SFA2), $n$-nonacosan-10-ol (SFA3), $n$-nonacosan-10,13-diol (SFA4), and $n$-nonacosan-5,10-diol (SFA5), identified in this study. TMS derivatives of these compounds were reported previously (Miyazaki et al. 2019). Besides SFAs, sugar compounds (sucrose, trehalose, arabitol, and mannitol) were measured using the method described above.

In this study, water-insoluble organic carbon (WIOC) is defined as a parameter calculated from the difference between the measured mass of organic carbon (OC) and water-soluble organic carbon (WSOC). A total organic carbon (TOC) analyzer (Model TOC-L$_{CHP}$, Shimadzu) was used to determine the WSOC mass concentration. A filter cut of 19.63 cm$^2$ was extracted and then filtered with a disc filter (Millex-GV, 0.22 μm, Millipore, Billerica, MA, USA), followed by injection of dissolved OC in the extracts into the analyzer. The measured carbon is defined as the WSOC. The data presented here were corrected using blanks. The mass concentration of OC was measured using a Sunset Lab OC/EC analyzer. A filter punch of 1.54 cm$^2$ was used for OC analysis. After measuring the OC and WSOC, the mass concentrations of WIOC were defined as the difference between those of OC and WSOC (WIOC = OC − WSOC).



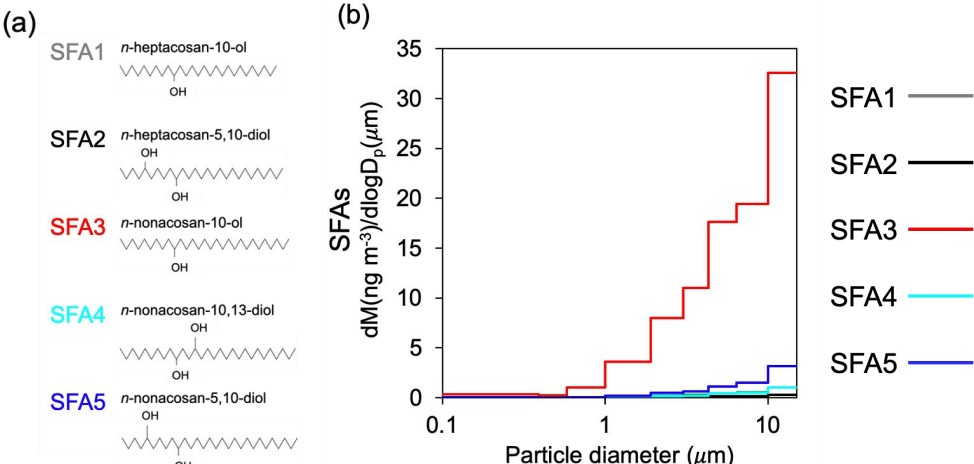

**Figure 2: (a) Molecular structures of secondary fatty alcohols (SFAs) identified in this study. *n*-heptacosan-10-ol, *n*-heptacosan-5,10-diol, *n*-nonacosan-10-ol, *n*-nonacosan-10,13-diol, *n*-nonacosan-5,10-diol are defined as SFA1, SFA2, SFA3, SFA4, and SFA5, respectively. (b) Mass size distributions of SFAs during the period between 21 and 28 May, 2010.**

## 3 Results and Discussion

### 3.1 Mass size distributions of secondary fatty alcohols in the aerosol samples

**Fig. 2b** shows the typical mass size distributions of the five identified SFAs obtained from samples collected in spring. SFA3 was the dominant SFA identified in all samples, followed by SFA5. **Table 1** summarizes the mass concentrations and

fractions of SFAs for each size range of the sample. The predominance of SFA3 among SFAs identified in aerosol samples was also reported previously at other forest sites (Miyazaki et al., 2019). Overall, the concentrations of SFAs were the highest in spring, which is the growing season (**Table 1**). The average concentrations of SFA3 ($10.2\pm10.3$ ng m$^{-3}$) and SFA5 ($0.52\pm0.69$ ng m$^{-3}$) in the bulk aerosol (the sum of submicrometre and supermicrometre aerosol mass) during spring are smaller than those in total suspended particulate (TSP) matter obtained at Tomakomai (TMK) experimental forest ($42°43′$ N,

$141°36′$ E) (Miyazaki et al., 2019).

In all seasons, the majority of the SFA mass resided in the supermicrometre size range. Specifically, the mass of *n*-nonacosan-10-ol in the supermicrometre size accounted for $93.4\pm7.1\%$, $84.2\pm17.8\%$, and $88.2\pm2.3\%$ of the total SFA mass in spring, summer, and autumn, respectively. In the following sections, the mass size distributions of *n*-nonacosan-10-ol are discussed as a representative SFAs.



**Table 1: Concentrations of secondary fatty alcohols (SFAs) in different size ranges of aerosol sample. ND indicates that the compound was not detected (i.e. below the lower detection limit).**

(a) *n*-nonacosan-10-ol (SFA3)

| Season | Sampling Period | Concentration (ng m$^{-3}$) | |
| --- | --- | --- | --- |
| | | Submicrometer ($D_p < 1.0$ μm) | Supermicrometer ($D_p > 1.0$ μm) |
| Spring | April 26–May 7, 2010 | 1.38 | 7.96 |
| | May 7–14, 2010 | 0.01 | 0.48 |
| | May 21–28, 2010 | 0.57 | 20.52 |
| Summer | June 22–29, 2009 | 0.82 | 14.91 |
| | July 8–16, 2010 | 0.11 | 1.73 |
| | August 6–13, 2010 | 0.90 | 1.57 |
| Autumn | October 6–13, 2009 | 0.68 | 5.24 |
| | October 13–21, 2009 | 1.35 | 8.17 |
| | October 21–28, 2009 | 0.45 | 4.27 |

(b) *n*-nonacosan-5,10-diol (SFA5)

| Season | Sampling Period | Concentration (ng m$^{-3}$) | |
| --- | --- | --- | --- |
| | | Submicrometer ($D_p < 1.0$ μm) | Supermicrometer ($D_p > 1.0$ μm) |
| Spring | April 26–May 7, 2010 | 0.04 | 0.29 |
| | May 7–14, 2010 | ND | 0.01 |
| | May 21–28, 2010 | 0.03 | 1.30 |
| Summer | June 22–29, 2009 | 0.03 | 0.78 |
| | July 8–16, 2010 | ND | 0.07 |
| | August 6–13, 2010 | 0.05 | 0.10 |
| Autumn | October 6–13, 2009 | 0.02 | 0.27 |
| | October 13–21, 2009 | 0.03 | 0.32 |
| | October 21–28, 2009 | 0.01 | 0.17 |

(c) *n*-heptacosan-10-diol (SFA1)

| Season | Sampling Period | Concentration (ng m$^{-3}$) | |
| --- | --- | --- | --- |
| | | Submicrometer ($D_p < 1.0$ μm) | Supermicrometer ($D_p > 1.0$ μm) |
| Spring | April 26–May 7, 2010 | 0.03 | 0.18 |
| | May 7–14, 2010 | ND | 0.06 |
| | May 21–28, 2010 | 0.02 | 0.49 |
| Summer | June 22–29, 2009 | ND | 0.33 |
| | July 8–16, 2010 | ND | 0.05 |
| | August 6–13, 2010 | 0.04 | 0.04 |
| Autumn | October 6–13, 2009 | ND | 0.24 |
| | October 13–21, 2009 | ND | 0.45 |
| | October 21–28, 2009 | ND | 0.25 |





(d) *n*-nonacosan-10,13-ol (SFA4)

| Season | Sampling Period | Concentration (ng m$^{-3}$) | |
| --- | --- | --- | --- |
| | | Submicrometer ($D_p < 1.0$ μm) | Supermicrometer ($D_p > 1.0$ μm) |
| Spring | April 26–May 7, 2010 | ND | 0.10 |
| | May 7–14, 2010 | ND | ND |
| | May 21–28, 2010 | ND | 0.44 |
| Summer | June 22–29, 2009 | ND | 0.18 |
| | July 8–16, 2010 | ND | 0.01 |
| | August 6–13, 2010 | 0.02 | ND |
| Autumn | October 6–13, 2009 | ND | 0.06 |
| | October 13–21, 2009 | ND | 0.09 |
| | October 21–28, 2009 | ND | 0.04 |

(e) *n*-heptacosan-5,10-diol (SFA2)

| Season | Sampling Period | Concentration (ng m$^{-3}$) | |
| --- | --- | --- | --- |
| | | Submicrometer ($D_p < 1.0$ μm) | Supermicrometer ($D_p > 1.0$ μm) |
| Spring | April 26–May 7, 2010 | ND | 0.02 |
| | May 7–14, 2010 | ND | ND |
| | May 21–28, 2010 | ND | 0.12 |
| Summer | June 22–29, 2009 | ND | 0.04 |
| | July 8–16, 2010 | ND | ND |
| | August 6–13, 2010 | ND | ND |
| Autumn | October 6–13, 2009 | ND | ND |
| | October 13–21, 2009 | ND | 0.00 |
| | October 21–28, 2009 | ND | 0.00 |

### 3.2 Seasonal changes in the mass size distributions of the secondary fatty alcohols

**Fig. 3** shows the average mass size distributions of SFA3 and SFA5 for each season. As described in the previous section, SFAs exhibited the highest concentrations in spring (the growing season). It is apparent that in spring, the peak diameter was greater than 10 μm. Interestingly, the peak diameter shifted to smaller sizes from spring to autumn, with the relative amount

10 of mass in the 1.9–3.0 μm diameter range becoming larger as the season progressed.

Because SFAs are compound groups of water-insoluble organic carbon (WIOC), their contribution to WIOC in terms of mass should be examined. The average mass concentration ratio of SFA3:WIOC in spring was 0.79%, whereas this period had most abundant SFA mass concentration among all the samples. Indeed, the $R^2$ value between *n*-nonacosan-10-ol and bulk WIOC concentrations was as high as 0.78 for that season (**Fig. 4b**). This result suggests that, although the mass fraction

15 of SFAs identified was small, a possible source of SFA is an important factor in controlling the WIOC mass in spring. The increase in the concentrations of WIOC with diameters greater than 10 μm was likely due to the influence of larger plant





debris in spring (Matsumoto et al., 2022), which has been regarded as a possible source of SFAs. However, this was not evident in other seasons when the mass concentrations were lower than those in spring.

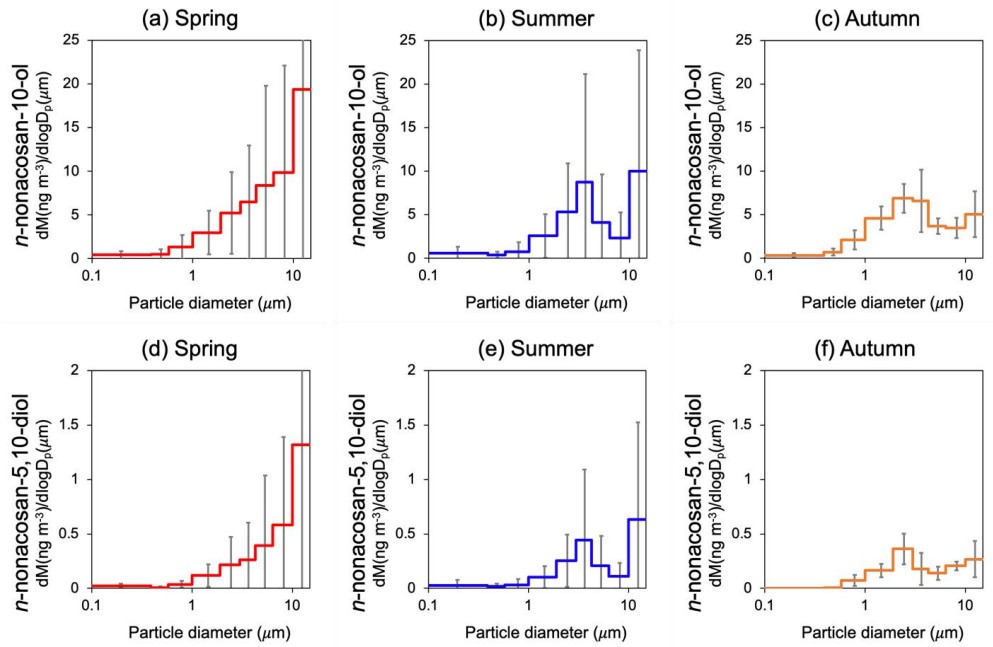

**Figure 3: The average mass size distributions of (a)(b)(c) *n*-nonacosan-10-ol (SFA3) and (d)(e)(f) *n*-nonacosan-5,10-diol (SFA5) in spring ((a) and (d)), summer ((b) and (e)), and autumn ((c) and (f)). Spring: 26 April to 7 May, 2010, 7 to 14 May 14, 2010 and 21 to 28 May, 2010; Summer, 22 to 29 June, 2009, 8 to 16 July, 2010 and 6 to August 13, 2010; Autumn, 6 to 13 October, 2009, 13 to 21 October, 2009 and 21 to 28 October, 2009.**

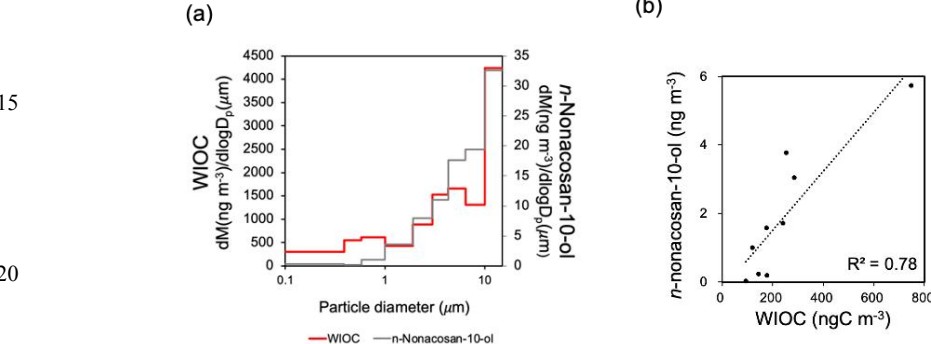

**Figure 4: (a) Mass size distribution of bulk water-insoluble organic carbon (WIOC) compared with that of *n*-nonacosan-10-ol during the period between 21 and 28 May, 2010. (b) A scatterplot of the mass concentrations between n-nonacosan-10-ol and bulk WIOC in spring (21–28 May, 2010).**





### 3.3 Possible origin and formation of secondary fatty alcohols in the aerosol

In this section, the origin and processes of formation of SFAs in the observed aerosols are described from the viewpoint of mass size distributions and seasonal differences in abundance. As indicated by the dominant mass fraction in the supermicrometre size range, SFAs in the observed aerosols were most likely PBAPs that are emitted into the atmosphere by

the local surface wind (Tegen et al., 2018). To support the discussion on the origin of PBAPs in relation to other PBAPs, correlations between the size-segregated mass concentrations of SFAs and several biogenic tracers were examined for each season.

### 3.3.1 Spring and summer

**Fig. 5** shows the mass size distributions of sucrose and trehalose compared with those of $n$-nonacosan-10-ol in each season.

Sucrose is the most abundant component of airborne pollen grains (Zhu et al., 2015) and was used as a pollen tracer compound. As expected, sucrose showed the largest mass concentration in spring, with the peak diameter ($D_p$) in the range of 4.3 to 6.4 m (**Fig. 5a**), whereas peak diameters (i.e. the shape of size distributions) were different between $n$-nonacosan-10-ol and sucrose in spring as well as in summer. Correlations between $n$-nonacosan-10-ol and sucrose concentrations were not apparent during these two seasons (**Fig. 6a**). In contrast to $n$-nonacosan-10-ol and sucrose, trehalose had the lowest mass

concentration in spring (**Fig. 5d–5f**). Trehalose is the most abundant sugar in soil (Jia and Fraser, 2011), and its contribution to soil dust has been previously reported (e.g., Jia et al., 2010; Medeiros et al., 2006). These studies suggest that trehalose can be used as a tracer for surface soil in atmospheric aerosols. The concentrations of $n$-nonacosan-10-ol did not show any significant correlation with trehalose concentrations in spring and summer (**Fig. 6b**).


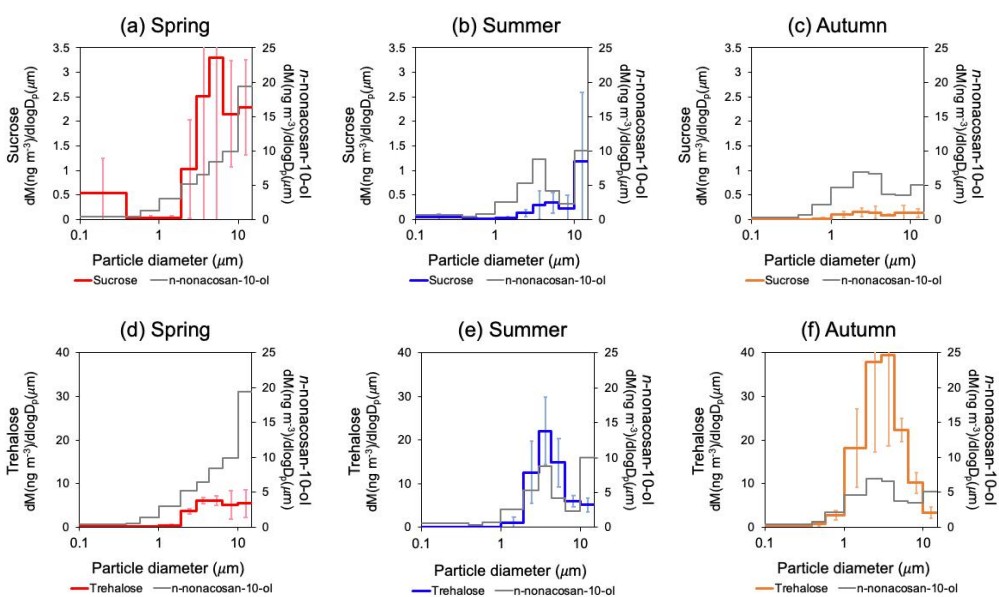

**Figure 5: The average mass size distributions of (a)(b)(c) sucrose and (d)(e)(f) trehalose, in spring, 26 April to 28 May, 2010**

**((a) and (d)), summer, 22 June to 13 August, 2010 ((b) and (e)), and autumn, 6 to 28 October, 2009 ((c) and (f)). Each size distribution was compared with that of $n$-nonacosan-10-ol during each corresponding period (gray).**





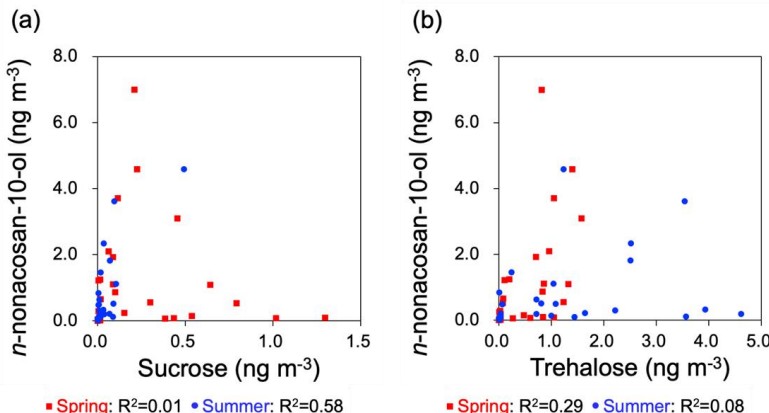

**Figure 6: Scatterplots of the mass concentrations between (a) *n*-nonacosan-10-ol and sucrose, and (b) *n*-nonacosan-10-ol and trehalose, in spring and summer.**

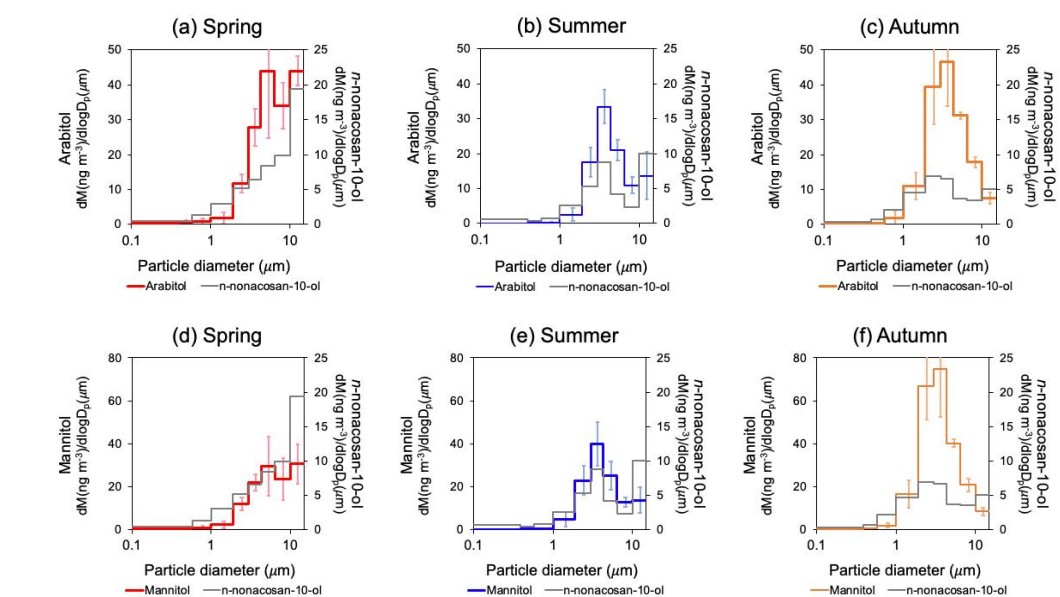

**Figure 7: The average mass size distributions of (a)(b)(c) arabitol and (d)(e)(f) mannitol in spring, 26 April to 28 May, 2010 ((a) and (d)), summer, 22 June to 13 August, 2010 ((b) and (e)), and autumn, 6 to 28 October, 2009 ((c) and (f)). Each size distribution was compared with that of *n*-nonacosan-10-ol during each corresponding period (gray).**

**Fig. 7** shows a comparison of the mass size distributions of *n*-nonacosan-10-ol with those of arabitol and mannitol during each season. Arabitol and mannitol have been used as tracers for airborne fungal spores (Fu et al., 2012). The size distribution for these two tracers was similar but was dissimilar to that of SFAs in each season—the mass concentrations showed insignificant correlations with those of *n*-nonacosan-10-ol in spring and summer ($R^2 < 0.25$) (**Fig. 8**). These insignificant correlations suggest that fungal spores are not the dominant source of *n*-nonacosan-10-ol during the spring and summer. In summary, the size distribution of *n*-nonacosan-10-ol did not show any significant correlation with those of





known biogenic tracers in spring, when SFAs had the maximum concentrations. Insignificant correlations between *n*-nonacosan-10-ol and these tracers were also observed in the summer. Overall, the results indicate that the majority of SFAs did not originate from pollen, soil, or fungal spores during spring and summer.

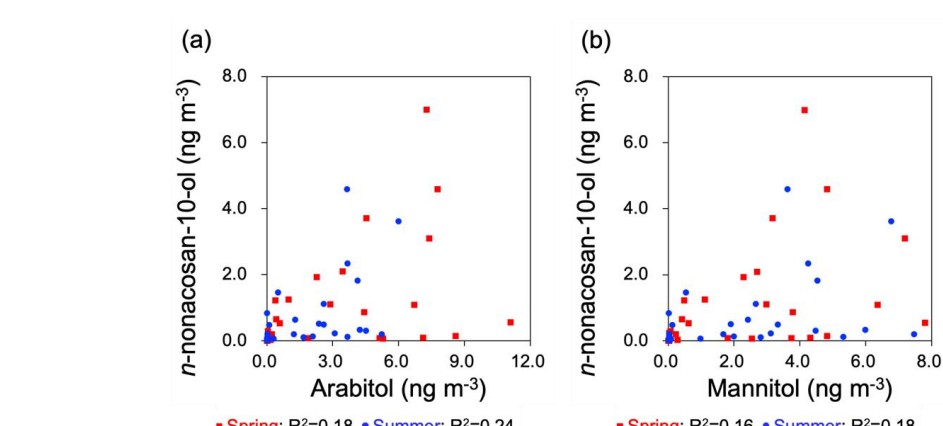

**Figure 8: Scatterplots of the mass concentrations between (a) *n*-nonacosan-10-ol and arabitol, and (b) *n*-nonacosan-10-ol and mannitol, in spring and summer.**

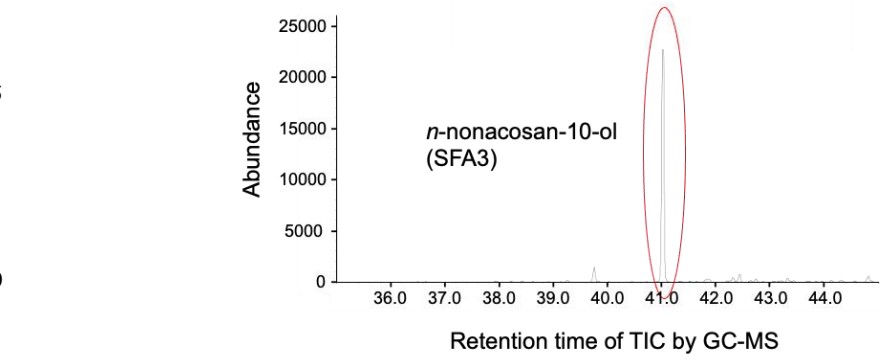

**Figure 9: GC-MS total ion chromatogram obtained for the trimethyl silyl (TMS) derivatives from leaf samples.**

25

As the most probable source of SFAs in the observed aerosols, *n*-nonacosan-10-ol is recognized as a major compound in epicuticular waxes found in gymnosperm species (e.g. Tulloch, 1976; Schulten et al., 1986; Sharma et al., 2018), such as *Sequoiadendron giganteum* (Yamamoto et al., 2008). Vioque and Kolattukudy (1997) measured secondary fatty diols in 30 other plant species, such as *Pisum sativum*. The main production of SFAs begins with the reduction of fatty acids through the formation of aldehydes as intermediates to produce FAs, which can form waxes in plants (Mudge, 2005). The wax-homologous series of long-chain lipids, including nonacosan-10-ol, are predominant (Gülz et al., 1994). In the aerosol samples, the carbon numbers of the identified SFAs were 27 and 29 (odd numbers). Indeed, SFAs with odd carbon numbers





can be produced by decarbonylation of FAs in plant leaves (Gülz et al., 1994). In order to confirm SFAs in a plant leaf which is representative of the sampling region, leaf samples of a Norway spruce (*Picea abies*) were collected at the campus of Hokkaido University nearby sampling site. **Fig. 9** presents an example GC-MS total ion chromatogram (TIC) of the TMS extract the leaf sample. The most abundant SFA, *n*-nonacosan-10-ol, found in the aerosol samples was indeed detected in the

leaf, at an average mass concentration of 46.4±11.3 mg g$^{-1}$. This result, together with the literature, supports the idea that the majority of the SFAs in the aerosols measured in this study originated from plant wax.

The strength of atmospheric emissions of SFAs may partly depend on the degree of epicuticular wax accumulation, which depends on the age of the leaf linked with phenology. The generation and regeneration of wax in plants occurs mostly in spring (Mohammadian et al., 2007). This explains the current results of the maximum concentrations of SFAs in the growing

season, which is attributable to a possible increase in the generation of plant wax. Waxes can be aerosolized from living plants and can be transported by wind (Nelson et al., 2018); thus, the most plausible process of formation of SFA aerosol particles is wind-driven emission from the leaf surface.

### 3.3.2 Autumn

The average mass concentrations of *n*-nonacosan-10-ol in autumn and summer were similar (**Table 1**). In contrast, trehalose

and mannitol had the highest concentrations in autumn (**Fig. 5** and **7**). The masses of these tracer compounds were mostly in the supermicrometre range. Similarly, the mass of arabitol was mostly in the supermicrometre size range, with a concentration greater than that in summer and the $R^2$ between *n*-nonacosan-10-ol and arabitol concentrations being 0.24. The $R^2$ values of concentrations of *n*-nonacosan-10-ol with those of trehalose and mannitol were 0.64 and 0.50 ($p < 0.01$ for both), respectively. However, these relationships are probably a coincidence because of the different seasonal changes in the mass

concentrations of these tracer compounds and *n*-nonacosan-10-ol, as well as the differences in the possible sources discussed in the previous section.

The concentration of *n*-nonacosan-10-ol positively correlated with that of sucrose in autumn (**Fig. 10a**). The flowering periods of the vegetation at the study site differed, and the possible sources of pollen were diverse. The two major types of vegetation at the study site are *Betula platyphylla* and *Quercus crispula Blume* (Nakai et al., 2003), the flowering periods of

which are from April to May and June to August, respectively. On the forest floor, *Sasa kurilensis* flowers were present at the study site in autumn. Particularly during the flowering period, when large atmospheric emissions of pollen are expected, pollen are often ejected in clumps that stick to nearby vegetation and are blown away after drying (Jones and Harrison, 2004). The positive correlation between *n*-nonacosan-10-ol and sucrose in autumn can be partly explained by the adhesion of pollen to plants (Visez et al., 2020), followed by atmospheric emissions together with *n*-nonacosan-10-ol from plant wax.

The predominance of the smaller peaks ($D_p$ = 1.9–3.0 μm) in the size distributions of *n*-nonacosan-10-ol in autumn compared with that in the other seasons (**Fig. 11**) can be partly explained by leaf senescence. The crystal transformation and degradation of epicuticular wax depend on the age of the leaf, which is linked to plant phenology (Mohammadian et al., 2007). Modification of epicuticular structures occurs during the aging of leaf surfaces, when epicuticular wax starts to degrade in autumn (Reicosky and Hanover, 1978). As a result, senescing leaves have lesser epicuticular wax than mature

leaves (Cao et al., 2013). The relatively large contribution of the smaller peak to the mass of *n*-nonacosan-10-ol is likely due to wax degradation. In summary, the major source of SFAs in the atmospheric aerosol samples collected in this study was probably plant wax; this notion is supported by the data for SFAs measured in the leaf samples and surrounding vegetation at the sampling site, and those in the literature. Further studies are required to measure SFAs in other types of plant leaves simultaneously with aerosol samples in a forest environment. The current findings on SFAs in aerosols provide new insights

into the origins and possible effects of PBAPs. Specifically, this study suggests that the aging processes of plants can affect the mass size distributions of SFAs and might thus alter the IN activity (Zhang et al., 2021).




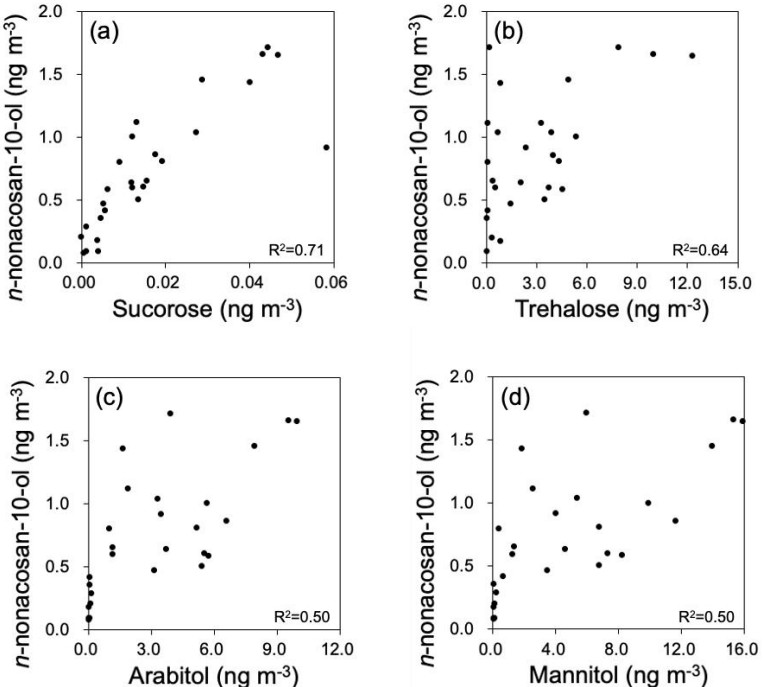

Figure 10: Scatterplots of the mass concentrations of *n*-nonacosan-10-ol vs. those of (a) sucrose, (b) trehalose, (c) arabitol, and (d) mannitol in autumn.

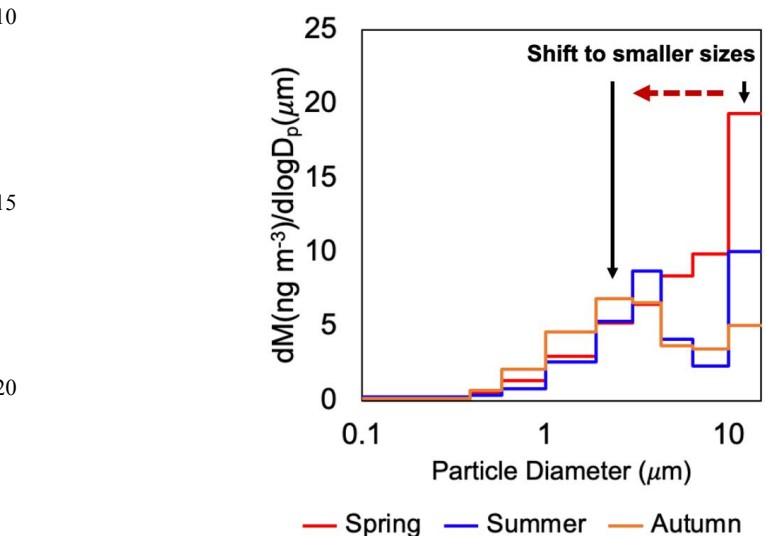

Figure 11: Seasonal changes in the average mass size distribution of *n*-nonacosan-10-ol.



## 4 Conclusions

In this study, we investigated the origin and formation of SFAs in atmospheric aerosols based on their mass size distribution in aerosol samples obtained from a deciduous forest site. Five SFAs were identified and their mass size distributions in the forest environment were measured for the first time. The mass concentrations of SFAs in the aerosols were the highest in the growing season (spring), which supports the results of a previous study at other forest sites. *n*-nonacosan-10-ol (SFA3) was the most abundant compound among the SFAs. On average, the mass of *n*-nonacosan-10-ol was mostly (>85%) in the supermicrometre size range, whereas the peak of the size range shifted to smaller sizes as the season progressed.

In spring, the mass size distributions were highly correlated with those of bulk WIOC, indicating that *n*-nonacosan-10-ol is an important factor controlling the WIOC mass. In spring and summer, the size distribution of *n*-nonacosan-10-ol was not significantly correlated with that of known biogenic tracers, such as pollen (sucrose), soil (trehalose), and fungal spores (arabitol and mannitol). Furthermore, *n*-nonacosan-10-ol was detected in Norway spruce leaf samples. The results of the aerosol and leaf samples, together with the literature on SFAs in plant leaves, suggest that SFAs mainly originated from plant wax. Moreover, the current results indicate that *n*-nonacosan-10-ol identified in this study can act as a possible new tracer for PBAPs.

In autumn, the mass peak of SFAs in the particle size greater than 10 μm, as determined in spring, was not apparent, whereas the mass in the smaller size range of 1.9–3.0 μm was relatively more compared with the larger size. The relatively large contribution of the smaller peaks to the size distribution of *n*-nonacosan-10-ol in autumn can be partly explained by leaf senescence, followed by the degradation of plant wax and subsequent atmospheric emissions. This study provides new insights into possible sources of PBAPs, namely, SFAs. Furthermore, our results suggest that the different growth stages of plants can result in differences in the size distributions of PBAPs emitted into the atmosphere, which may affect the IN properties of aerosol particles.

## Data availability

The measurement data for the aerosol samples are provided in the Supplementary Material. All other data are available upon request.



**Author contributions**

YC and YM designed the study and wrote the manuscript. YC performed the GC-MS experiments. YM performed the aerosol sampling. KK managed the sampler. YC, YM, and ET analyzed the data.

**Competing interests**

The authors declare that they have no conflict of interest.

**Acknowledgements**

We thank K. Yamanoi for his managing the sampling site and for his help with the aerosol sampling.

**Financial supports**

This research was supported by Grants-in-Aid for Scientific Research (B) (16H02931) from the Ministry of Education,
Culture, Sports, Science, and Technology (MEXT), Japan, and by JST SPRING (Grant Number JPMJSP2119) via the DX Doctoral Fellowship of Hokkaido University.

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
