# Peer review of "Origin of secondary fatty alcohols in atmospheric aerosols in a cooltemperate forest based on their mass size distributions"

_Biogeosciences, 2023_

## Author Comment (AC1)

**Responses to the comments of Referee#1**

General comments:

Manuscript entitled "Origin of secondary fatty alcohols in atmospheric aerosols in a cool-temperate forest based on their mass size distributions" to investigate the origin and formation of secondary fatty alcohols (SFAs) based on their mass size distribution in aerosol samples obtained from a deciduous forest site. This manuscript gives us a clearer understanding of the five SFAs and their mass size distributions in the forest environment for the first time. The author concluded that the mass concentrations of SFAs were the highest in spring and the mass size distributions were highly correlated with bulk WIOC. Furthermore, SFAs were induced that mainly originated from plant wax from the results of the aerosol and leaf samples. This study provides new insights into possible source of primary biological aerosol particles (PBAPs) and their effects on the ice nucleation activity of aerosol based on seasonal changes in particles. Finally, these results demonstrated that the different growth stages of plants can result in differences in the size distributions of PBAPs. The Reviewer supports this manuscript for publication in *Biogeosciences* if the authors could carefully address the following comments.

**Reply: We appreciate the referee's valuable comments on our work. Our responses to the specific comments and details of the changes made to the manuscript are given below.**

Specific comments:

1. Why did the authors analyze only spring, summer, and autumn atmospheric samples, but not winter samples? In my opinion, the winter sample can be used as a blank comparative analysis, excluding the interference of other environmental factors. Please explain it in detail.

**Reply 1: Our previous study on source apportionment of organic aerosols at the same forest site suggested that majority of aerosols originated from the local forest in spring, summer, and autumn (Miyazaki et al., 2012). On the other hand, the observed aerosols in winter were significantly affected by marine and**

anthropogenic sources rather than showing blank or background levels at this forest site (Miyazaki et al., 2012). Because we focused on effects of forest vegetation on SFAs in aerosols, we have not shown the data in winter in this study. To clearly mention this, we made additional statement as follows:

**P.2 L. 30:** *"Indeed, the source apportionment study suggested that majority of aerosols originated from the local forest in spring, summer, and autumn, whereas the observed aerosols in winter were significantly affected by marine and anthropogenic sources (Miyazaki et al., 2012). Because we focus on effects of forest vegetation on SFAs in aerosols, the data obtained in spring, summer, and autumn are shown and discussed in this study."*

2. Page 2 Lines 32-33: Why is the particle size only analyzed to 10 μm? This is very confusing, 10 μm is represent the maximum particle size or larger than 10 μm? How is the particle size measured? If it is based on the sampler, you may be able to try more levels of impactors.

**Reply 2: The top stage of the impactor collected particles with diameter larger than 10 μm ($D_p$ > 10 μm) as we described in the original text. The particle size cut was based on theoretical calculation which was reflected to the structure of impactor. In general, atmospheric residence time of larger particles with diameters of tens of micrometer is very short (e.g., an order of mins to tens of mins), which may not have significant impact on atmospheric chemical field. We believe that the data shown in our study can be a useful reference for further studies on aerosol SFAs.**

3. Humidity can influence the particle size distribution. Did the authors consider environmental factors (such as humidity) in the analysis.

**Reply 3: First of all, the sampling was made without temperature or humidity control in this study. We agree that relative humidity can influence the particle size distribution. However, SFAs are water-insoluble fractions of organic aerosol which might be less hygroscopic, and their size distributions may be less affected by relative humidity. Indeed, in our study, the size distributions of SFAs did not necessarily show the peak in the larger size ranges in summer, when relative humidity is highest in all the seasons at the site. Rather, our data suggested that**

the mass concentrations and their maximum size range was more influenced by the plant activities or age as we discussed in the manuscript. As the referee pointed out, the following description has been added in the revised manuscript:

**P.11 L. 36:** *"It is noted that relative humidity can influence particle size distribution. However, SFAs are water-insoluble fractions of organic aerosol (Fig.4) which might be less hygroscopic, and their size distributions may be less affected by relative humidity. Indeed, the size distributions of SFAs observed in this study exhibited peaks in the larger size ranges in spring, despite the highest relative humidity was observed in summer at the site, suggesting insignificant effects of relative humidity on the size distributions in this study."*

4. Page 2 Line 34-36: For the analysis of SFAs, the authors choose the sampling time of 1 week and at a flow rate of 120 L min$^{-1}$, this indicating that the concentration of SFAs is lower. So please explain the atmospheric significance of analyzing SFAs and its seasonal variations.

**Reply 4: As we described in the *introduction* section, the seasonal variations in the mass concentration and size distributions of SFA provide important clues for understanding its origin. Because SFAs were measured as "molecular tracers" of origin of aerosols, the low concentration does not mean that it is not important. Rather, SFAs reflect the changes of the activity of local vegetation in different seasons as shown in this study. Our study suggested that SFAs originated from plant waxes, the emission strength of which is likely controlled by the seasonal difference of plant growth. The significance of analyzing SFAs is that such new tracers of PBAPs provides new insight into biogenic sources of organic aerosols and their impact on ice nuclei activity.**

5. Page 3 Lines 16-17: "a quarter of each sample filter with an aera of 10.18 cm2 was extracted using a mixture of dichloromethane and methanol (2:1, V:V)". Why did the author choose the mixture of dichloromethane and methanol (2:1) as the extracted solution?

**Reply 5: The method using mixture of dichloromethane and methanol has been already established as an optimal solvent for the extraction of organic compounds in aerosol samples. We adopted the optimal extraction method used in many of**

previous studies (Fu et al. (2009) and many of other relevant studies). Furthermore, Miyazaki. (2019) successfully extracted and measured five compounds of SFAs in aerosol samples obtained at forest sites. Therefore, we also used this optimal method in this study.

6. Page 7 Figure 3: Why is the error bar so large for the spring and summer samples in Figure 3?

**Reply 6: Three sets of samples were used for the calculation of average values in each season. In spring and summer, concentrations in one sample set were significantly different from those in the other two sample sets, which resulted in the large error bars despite the similar size distributions among the three sets. One possible explanation is that precipitation that occurs in summer and spring (Miyazaki et al., 2012) caused decrease in the concentrations of aerosols. Because the shape of size distributions of aerosol SFAs was similar in each season, we believe that this does not significantly affect our conclusions. To clarify the statement, we have added the sentence as follows:**

**P.6 L. 10: "*The large standard deviation shown in Fig. 3 can be partly explained by the effect of precipitation that occurred in spring and summer (Miyazaki et al., 2012), which might cause a decrease in the concentrations of SFAs in aerosols.*"**

7. Page 9 Figure 7: In the seasonal variation shown in Figure 7, the mass size distribution does not change significantly compared with Figure 11.

**Reply 7: Figure 7 shows the seasonal changes in the size distributions of two molecular tracers of PBAPs in color, which are different from those of *n*-nonacosan-10-ol summarized in Figure 11. The difference and similarity in the size distributions between Figure 7 and Figure 11 are discussed in the section 3.3 as well as shown in the original Figure 10.**

8. Page 12 Figure 10: It is suggested that the author place Figure 10 into the supplementary material.

**Reply 8: As the referee suggested, Figure 10 has been moved to the supplementary material in the revised manuscript.**

**References**

Fu, P, Kawamura, K., Chen, J., and Barrie, L.: Isoprene, monoterpene, and sesquiterpene oxidation products in the high Arctic aerosols during late winter to early summer, *Environ. Sci. Technol.*, 43, 4022–4028, 2009.

Miyazaki, Y., Fu, P., Kawamura, K., Mizoguchi, Y., and Yamanoi, K.: Seasonal variations of stable carbon isotopic composition and biogenic tracer compounds of water-soluble organic aerosols in a deciduous forest, *Atmos. Chem. Phys.*, 12, 1367-1376, doi:10.5194/acp-12-1367-2012, 2012.

Miyazaki, Y., Gowda, D., Tachibana, E., Takahashi, Y., and Hiura, T.: Identification of secondary fatty alcohols in atmospheric aerosols in temperate forests, *Biogeosciences*, 16, 2181-2188, https://doi.org/10.5194/bg-16-2181-2019, 2019.

---

## Author Comment (AC2)

**Responses to the comments of Referee#2**

General comments:

This paper presents the atmospheric loadings and seasonal distributions of secondary fatty alcohols in size-resolved aerosols from a deciduous forest canopy at Hokkaido, Japan. The results obtained have been interpreted logically. Based on the distributions of n-nonacosan-10-ol and its non-linear relations with the primary organic aerosol markers in the same samples such as sucrose, trehalose, mannitol and arabitol in spring, summer and autumn together with the comparisons of the SFAs measured in plant leaves, authors have found that the SFAs are derived from plant waxes. Overall, the data presented here and the conclusions drawn from them are interesting and make a substantial contribution to the community of atmospheric- and biogeo-sciences. Therefore, I recommend this paper for final publication in BG, after addressing the following comments.

**Reply: We appreciate the referee's valuable comments on our work. Our responses to the specific comments and details of the changes made to the manuscript are given below.**

Specific comments:

1. Section 3.1 - Page 4, Lines 12-15: The average concentrations of SFA3 --- are smaller than those --- (Miyazaki et al., 2019). Here it is important to note the literature values and describe the potential reasons behind such lower levels (10~20 times) compared to those observed at Tomakomai experimental forest.

**Reply 1: In total suspended particulate (TSP) matter obtained previously at Tomakomai (TMK) experimental forest (42°43′ N, 141°36′ E), the average concentrations of *n*-nonacosan-10-ol (SFA3) and *n*-nonacosan-5,10-diol (SFA5) were ~100 ng m$^{-3}$ and ~8 ng m$^{-3}$, respectively, in spring (Miyazaki et al., 2019). These concentrations are 10–15 times larger than those at this study site (42°59′ N, 141°23′ E). The difference in the concentration levels is likely due to several factors: the area of TMK experimental forest (2715 ha) is substantially larger than that (147 ha) of the forest site in this study. Moreover, leaf area index (LAI) of TMK**

(~4–6) (Hiura, 2001) is larger than that of this study site (<4) (Nakai et al., 2003). These suggest that the TMK experimental forest has more biomass, which has a potential to emit SFAs, than that in the current study site. Furthermore, TMK consists of a wide variety of vegetation including mature and secondary deciduous trees and man-made coniferous trees, the number of which is much larger than that of this study site. This implies that TMK has much more potential to emit PBAPs, although dominant vegetation species that emit SFAs has not been specifically identified. Taking account of the referee's comment, we made additional statement with the values in literature as follows:

Page 4, Lines 12: *"The average concentrations of SFA3 (10.2±10.3 ng m$^{-3}$) and SFA5 (0.52±0.69 ng m$^{-3}$) in the bulk aerosol (the sum of submicrometre and supermicrometre aerosol mass) during spring are smaller than those (~100 ng m$^{-3}$ and ~8 ng m$^{-3}$, respectively) in total suspended particulate (TSP) matter obtained at Tomakomai (TMK) experimental forest (42°43' N, 141°36' E) (Miyazaki et al., 2019). If most of these SFAs were originated from forest vegetation, the difference in the concentration levels is attributable to the larger area ( 2715 ha) and leaf area index (LAI; ~4–6) with much more plant species in the TMK experimental forest (Hiura, 2001) compared to those (150 ha and <4, respectively) of the current research forest site (Nakai et al., 2003). "*

2. Section 3.3.1 – Page 8, Lines 13-14: Correlations between --- two seasons (Fig. 6a). It is contrary to a significant correlation (R2 = 0.70) reported between the *n*-nonacosan-10-ol and sucrose in forest aerosols by Miyazaki et al., 2019. So, I suggest the authors to compare the data (the sum of submicrometre and supermicrometre aerosol mass) with that (in TSP) reported by Miyazaki et al., 2019 in order to make it clear.

Reply 2: Although the correlation coefficient between *n*-nonacosan-10-ol and sucrose was discussed in both studies, time scales and size bins of data used for the calculation in the two studies are completely different. The significant correlation shown in Miyazaki et al. (2019) was based on the time series data of TSP on a time scale of one year. The correlation shown in their study meant that the seasonal trends of the SFA and sucrose concentrations in TSP were similar, both of which showed peaks in the growing season. On the other hand, the relation between *n*-nonacosan-10-ol and sucrose in our study was based on the data of each size bins obtained in one week of each season (spring or summer). The insignificant

correlation meant that the size distributions with peaks were different between the two compounds, suggesting the different sources. Although it is difficult to calculate correlation coefficient for TSP in this study because of the limited number of data (three in one season), the seasonal trend of the *n*-nonacosan-10-ol and sucrose concentrations in TSP in our study were similar to that reported in Miyazaki et al. (2019) (i.e., maximum in spring).

**References:**

Hiura, T.: Stochasticity of species assemblage of canopy trees and understory plants in a temperate secondary forest created by major disturbances, *Ecol. Res.*, 16, 887–893, 2001.

Miyazaki, Y., Gowda, D., Tachibana, E., Takahashi, Y., and Hiura, T.: Identification of secondary fatty alcohols in atmospheric aerosols in temperate forests, *Biogeosciences*, 16, 2181-2188, https://doi.org/10.5194/bg-16-2181-2019, 2019.

Nakai, Y., Kitamura, K., Suzuki, S., and Abe, S.:Year-long carbon dioxide exchange above a broadleaf deciduous forest in Sapporo, Northern Japan, *Tellus*, 55B, 305–312, 2003.

---

## Author Response (AR2)

**Responses to the comments of co-editor-in-chief**

**Comments:**

1. p.2, l30: add 'the' before 'majority: ... suggested that the majority of aerosols.

2. p.4, l. 17: Remove 'were' in 'were originated from'.

**Reply: We appreciate the co-editor-in-chief's comments on technical errors We have carefully checked the manuscript and modified the words including above.**